# *Treponema pallidum* among Female Sex Workers: A Cross-Sectional Study Conducted in Three Major Cities in Northern Brazil

**DOI:** 10.3390/pathogens10080923

**Published:** 2021-07-22

**Authors:** Luiz Fernando Almeida Machado, Jacqueline Cortinhas Monteiro, Leonardo Quintão Siravenha, Marcelo Pereira Mota, Marlinda de Carvalho Souza, Adalto Sampaio dos Santos, Márcio Ronaldo Chagas Moreira, Rogério Valois Laurentino, Aldemir Branco Oliveira-Filho, Maria Alice Freitas Queiroz, Sandra Souza Lima, Ricardo Ishak, Marluísa de Oliveira Guimarães Ishak

**Affiliations:** 1Programa de Pós-Graduação em Biologia de Agentes Infecciosos e Parasitários, Instituto de Ciências Biológicas, Universidade Federal do Pará, Belém 66075-110, PA, Brazil; alicefgarcia@gmail.com (M.A.F.Q.); rishak@ufpa.br (R.I.); marluisa.malu@gmail.com (M.d.O.G.I.); 2Laboratório de Virologia, Instituto de Ciências Biológicas, Universidade Federal do Pará, Belém 66075-110, PA, Brazil; jacqueline@ufpa.br (J.C.M.); leo.siravenha@yahoo.com.br (L.Q.S.); mpm.biomedic@gmail.com (M.P.M.); valois@ufpa.br (R.V.L.); saraujo@ufpa.br (S.S.L.); 3Centro de Hematologia e Hemoterapia do Acre, Rio Branco 69900-607, AC, Brazil; marlinda06@gmail.com (M.d.C.S.); adautobiovida@yahoo.com.br (A.S.d.S.); 4Seção de Virologia, Laboratório Central de Saúde Pública do Amapá, Macapá 68905-320, AP, Brazil; mormarcio@hotmail.com; 5Grupo de Estudo e Pesquisa em Populações Vulneráveis, Instituto de Estudos Costeiros, Universidade Federal do Pará, Bragança 68600-000, PA, Brazil; olivfilho@ufpa.br

**Keywords:** epidemiology, public health, *Treponema pallidum*, female sex workers, vulnerability, health promotion, Brazil

## Abstract

Female sex workers (FSWs) are an important group of people vulnerable to sexually transmitted infections. Northern Brazil is a rural and socioeconomically underdeveloped region, with lack of epidemiological information on syphilis in key populations. This study investigated the prevalence and factors associated with exposure to *Treponema pallidum* among FSWs in three major cities in northern Brazil. This cross-sectional study was conducted with a convenience sample of 415 FSWs from the cities of Belém, Macapá, and Rio Branco. Blood samples and personal data were collected from January 2009 to August 2010. Rapid plasma reagin and immunoenzymatic assays were used to detect antibodies against *T. pallidum*. Logistic regression models were used to determine factors associated with exposure to *T. pallidum*. In total, 36.1% were exposed to *T. pallidum*, and 15.7% had active syphilis. Sexual risk behaviors, use of illicit drugs, low education, and reduced monthly income were associated with exposure to *T. pallidum*. The high rate of exposure to *T. pallidum* indicates the urgent need for measures to identify, treat, and prevent syphilis and an education program for the valuation, care, and social inclusion of FSWs in northern Brazil.

## 1. Introduction

Syphilis is a sexually transmitted infection (STI) caused by *Treponema pallidum* subspecies *pallidum* (*T. pallidum*), an invasive pathogen that may cause systemic disease when not treated properly, resulting in long-term cardiac and neurological complications [1]. The bacterium is transmitted mainly through sexual intercourse, causing initial lesions in genital organs, facilitating transmission of other infectious agents, including the human immunodeficiency virus [2,3]. Syphilis represents a serious public health problem worldwide, and the number of new cases has resurfaced with greater intensity in recent decades [4].

The World Health Organization (WHO) estimates the occurrence of new cases of syphilis worldwide at 6.3 million [5]. In Brazil, it was observed that a consistent increase in the number of cases may be related to an increase in testing coverage, but also to the declining availability of penicillin in the world, among other factors [6]. In 2018, the number of reported cases of acquired syphilis was 185,051, which represents a rate of detection of 75.8/100,000 inhabitants [7].

Eleven countries that reported the occurrence of syphilis showed more than 5% of female sex workers (FSWs) with an active infection in 2018 [8]. FSWs are of major importance as they live in a vulnerable social condition and frequently present a high behavior risk of sexual transmission associated with their daily activities, including multiple sexual partners, inconsistency in the usage of condoms, and difficulty in negotiating prevention strategies against STI with their clients [5,6,9,10].

Serological investigation of syphilis among FSWs from 28 countries showed a mean infection prevalence 2.3% [11]. In Brazil, the prevalence of syphilis among FSWs in 12 major Brazilian cities was 8.5% [12], but in Brazilian municipalities in the South Region, a prevalence of 19.7% has already been demonstrated [13]. Epidemiological aspects of syphilis among FSWs in northern Brazil are still poorly understood. In the state of Pará, the second largest in the North Region, high prevalence of syphilis was shown among FSWs who worked along highways (36.94%) in three countryside cities of the state of Pará (14.1%) and the Marajó Archipelago (41.1%) [14,15,16]. The presence of *T. pallidum* strains with the A2058G and A2059G resistance point mutations in the rRNA gene was also a common finding among FSWs in the Marajó Archipelago [16].

Despite the scarcity of epidemiological information on syphilis in northern Brazil, the few scientific records indicate a worrying scenario. The present study investigated the prevalence and factors associated with exposure to *T. pallidum* among FSWs who offered sexual services in three major cities in northern Brazil.

## 2. Results

### 2.1. Sample Characteristics

In the largest and most important cities in the Brazilian states of Pará (Belém, *n* = 21), Amapá (Macapá, *n* = 4), and Acre (Rio Branco, *n* = 3), 28 sex trade sites were identified and visited by the research team (Figure 1). At each location, at least half of the FSWs offering sexual services agreed to participate in this study. From January 2009 to August 2010, 648 FSWs were approached and invited to participate in this report, and 415 of them accepted (Belém, *n* = 360; Macapá, *n* = 31; Rio Branco, *n* = 24).

The sociodemographic and behavioral characteristics of FSWs are shown in Table 1. The average age was 35.1 years old. The oldest subjects were detected in the city of Belém, Pará (average age = 43.0 years), and the youngest, in the city of Rio Branco, Acre (average age = 25.1 years). Most women were single, had low education (≤8 years of study), had low monthly income (less than or equal to one Brazilian minimum wage), used noninjectable illicit drugs, used condoms during sexual intercourse, and maintained sexual relations with clients from other Brazilian states. Some FSWs also reported having sexual relations with non-Brazilian clients, mainly in the city of Rio Branco. The mean number of clients was 9.5 per week, the highest detected in the cities of Belém (10.3 clients/week) and Rio Branco (11.5 clients/week). There was a record of STI history among the subjects, with the highest frequencies in the cities of Amapá (15.4%) and Rio Branco (22.7%).

### 2.2. Exposure to T. pallidum and Active Syphilis

In total, 150 (36.1%) FSWs were exposed to *Treponema pallidum* in the three major cities in northern Brazil (Table 2). Most of them (20.5%) had serological results that indicated past infection. The rates of past infections were higher than the values detected for recent infections with *T. pallidum* in the cities of Macapá, Belém, and Rio Branco (Table 1). The overall prevalence of recent infections was 15.7% (Table 2). The highest rate was registered among FSWs in the city of Belém (16.7%), and the lowest among subjects in the city of Rio Branco (4.2%) (Table 1). In this study, nine (2.2%) FSWs showed positive results for RPR and negative results for ELISA. These cases were considered cross-reactions and were interpreted as false-positive results (i.e., FSWs also susceptible to infection with *T. pallidum*) (Table 2).

### 2.3. Factors of Exposure to T. pallidum

Logistic regression models were made, and seven factors were associated with exposure to *T. pallidum*: up to 8 years of study, low monthly income, use of illicit drugs, unprotected sex, anal sex, more than 10 sexual partners per week, and STI history (Table 3). The Hosmer–Lemeshow test showed that the final multivariate model (_HL_χ^2^ = 5.1; *p* = 0.4) had a good fit. Unprotected sex (adjusted OR = 6.4), anal sex (aOR = 4.6), and STI history (aOR = 7.1) were the main variables associated with exposure to *T. pallidum* (Table 3). The factors not associated with exposure to *T. pallidum* are listed in Appendix A.

## 3. Discussion

This report with a sample of FSWs working in cities in northern Brazil shows not only the relevance of this vulnerable population in the acquisition and spread of *T. pallidum* but also the situation of deprivation of women who cannot find any other possible solution to work and acquire resources—uneducated, poor, unable to claim for protected sex, having had to handle many clients, which will finally lead to contamination with STI, including *T. pallidum*. Their situation is not different from that of other FSWs residing elsewhere in the country. The presence of *T. pallidum* was described among FSWs residing in the capital cities of the states of Acre, Amapá, and Pará in northern Brazil. The high rates of infections (past and recent) indicate their vulnerability to *T. pallidum*, which must call the attention of health authorities within the region and elsewhere in Brazil.

Syphilis serology results are not always easy to interpret given the complex pathogenesis of the infection and the immune response of each individual. In the present study, we considered active syphilis (ongoing or recent infection) when the RPR result was ≥1:8; however, it is noteworthy that about 20% of cases of primary syphilis may have a negative RPR test in the presence of treponemal antibodies [17]. Regardless of any initial interpretation, in the case of suspected syphilis, all participants were properly treated. In addition, it may occur in individuals manifesting less than a fourfold decline or persistently positive RPR titers (serofast), which may represent treatment failure, reinfection, or satisfactory immune response [18].

In the present study, the prevalence of active syphilis (recent infections) among FSWs in three cities in the interior of Pará was 14.1% [15], in which Belem showed a prevalence of recent infections of 16.7% and of past infections of 19.7% by *T. pallidum*. An expanded seroepidemiological investigation in 10 municipalities and 18 riverside communities in northern Pará, including the Marajó Archipelago and women working along highways, detected a range of active syphilis of 36.9–41.1% and identified strains containing mutations of resistance to treatment and coinfections with hepatotropic viruses [14,16]. Continuous epidemiological studies with vulnerable people are needed in northern Brazil and elsewhere. Monitoring of FSWs in 12 major Brazilian cities showed a significant increase in active cases of syphilis from 2009 (1.4%) to 2016 (8.5%) and the urgent need to take appropriate preventive measures to protect this group and reduce the dispersion of this spirochete [12]. Although there is no previous information on syphilis among FSWs in the cities of Macapá and Rio Branco, the rate of this STI was similar to that found in other underdeveloped countries, including Peru (0.8%), Mexico (7.8%), Guatemala (1.3%), and Iran (0.4%) [19,20,21,22].

Sexual risk behaviors, use of illicit drugs, low education, and reduced monthly income were associated with exposure to *T. pallidum* among FSWs. Unprotected sexual relations, anal sex, and multiple sexual partners are commonly associated with exposure to *T. pallidum* among FSWs in Pará [14,15], in other areas of Brazil [23], and in several other countries, such as Colombia [24], China [25], and India [26]. Low perception of the risk of syphilis was associated with poor education, low monthly income, and use of psychotropic drugs (licit and illicit), which has been reported in Brazilian epidemiological studies, including in the state of Pará [14,15,16,27,28,29,30,31]. Although this study investigated only the presence of *T. pallidum*, other epidemiological reports conducted with FSWs in northern Brazil have already registered the presence of multiple infectious agents, which highlights the history of STI as a risk factor [16,32].

The results found indicate very limited support offered for the assistance and health promotion of FSWs. Active investigations should be stimulated to reach, identify, treat, and provide means of prevention of STI among FSWs not only in urban settings but also in riverside communities, small cities, and roads, among others. Community health workers can identify FSWs, provide condoms free of charge, guide them to seek diagnosis and treatment, and provide access to Brazilian primary health care [16]. Another necessary key intervention is the development of educational programs, not only to guarantee the health of the individual, but also to value and promote self-esteem, self-confidence, self-care, women’s rights, and social inclusion. These activities may modify these women’s perception of the risk of STI acquisition and facilitate their access to health services. These two important points, perception of the risk to STI and barriers to accessing existing health services, should be studied in the future. Overall, these interventions and complementary studies are essential to understand the epidemiological scenario and reduce the spread of *T. pallidum* and other pathogens among FSWs and, consequently, in the general population.

This study has limitations that should be considered. First, this study used convenience sampling. Second, the sample size of subjects was small in two cities: Macapá and Rio Branco. In addition, this study did not access FSWs who offered their services over a digital network, as already registered [30]. Lastly, the cross-sectional design of the study limits its capacity to establish causality.

## 4. Materials and Methods

### 4.1. Study Area

Northern Brazil, including a portion of the rainforest in the Amazon, is a rural and socioeconomically underdeveloped region, with high levels of poverty, limited transport infrastructure, and precarious health services. Along with this epidemiological scenario, poverty, low education, malnutrition, domestic violence and child abandonment, sex work, and illicit drug use and trafficking are common in the region [33]. The states of Acre, Amapá, and Pará show interesting epidemiological characteristics, such as large flow of people and products because of trade and tourism in northern Brazil, many municipalities with low or very low development index, health service structure with many limitations, many refugees from Haiti and Venezuela in the past decade, and border with other seven South American countries. The main cities of these states are Belém (1,499,641 inhabitants), Macapá (512,902 inhabitants), and Rio Branco (413,418 inhabitants), totaling approximately 2,425,000 inhabitants [34].

### 4.2. Sampling

This study was based on convenience sampling, which obtained blood samples and personal data from FSWs who offered their sexual services in major cities in the states of Pará (Belém, *n* = 360), Amapá (Macapá, *n* = 31), and Acre (Rio Branco, *n* = 24), northern Brazil (Figure 1). FSWs were contacted at their workplaces (streets, bars, and clubs), either during the day or at night, from January 2009 to August 2010. The study included women (biologically determined at birth) of any age who reported having sexual intercourse in exchange for money and resided in the cities for at least 3 months. Those women who did not agree to either blood sample collection or answering of a questionnaire were excluded.

### 4.3. Study Design

Initially, a survey of sites offering sexual services was conducted with residents of each city. After registering and identifying the locations and times of offering sexual services, a research team went to each location indicated to access and invite FSWs to participate in this study. FSWs were previously informed about the research objectives and invited to participate in the study without any payment of resources. Those who accepted to take part in the investigation answered an epidemiological form (during a face-to-face interview) [15]. This form included information regarding age, marital status, number of years of studies, monthly income (last 12 months), use of illicit drugs (injectable/noninjectable), condom use during sexual intercourse (last 30 days), number of sexual partners per week (last 30 days), anal sex (last 30 days), sexual intercourse with clients from other Brazilian states and/or other countries, and history of previous STI. Then, all the subjects provided written consent to participate in the study and had a blood sample collected by the research team.

### 4.4. Laboratory Tests

Peripheral blood samples (10 mL) were collected using a vacuum tube containing EDTA, an anticoagulant. Plasma was separated by centrifugation (10,000 rpm for 10 min) and stored at −20 °C in the Virus Laboratory of the Federal University of Pará (Belém, Brazil) until use. All samples were subjected to a nontreponemal screening test, rapid plasma reagin (RPR; RPR Brás, Laborclin, Brazil), and a treponemal test, enzyme-linked immunosorbent assay, for the detection of specific antibodies against *T. pallidum* (ELISA; ETI-Treponema Plus, DiaSorin, USA). All reactions were performed following the instructions of the manufacturers. Active syphilis (ongoing or recent infection) was defined as RPR result ≥1:8 [23]. Negative RPR reactions and positive ELISA tests were considered past infections with *T. pallidum*. Recent and past infections were considered exposure to *T. pallidum*.

### 4.5. Statistical Analysis

Study data were stored in an Excel database (Microsoft Corp., Redmond, WA, USA) and converted to SPSS (IBM, Armonk, NY, USA). Confidence intervals (95% CI) were calculated for the prevalence of syphilis in each city and overall. Chi-square tests were used to compare variables and to determine *p*-values. OR and 95% CI were used as measures of the strength of association between exposure to *T. pallidum* (outcome) and independent variables. Variables associated with the outcome, with *p* < 0.20 in the bivariate analysis, were inserted in a backward stepwise logistic regression model (multivariate analysis). The fit of the final model was assessed using the Hosmer–Lemeshow goodness-of-fit test. A *p*-value of <0.05 significance value was considered for all analyses using SPSS 23.0 for Windows.

## 5. Conclusions

High rates of infections (past and recent) with *T. pallidum* were detected among FSWs in the cities of Belém, Macapá, and Rio Branco. Actions related to the supply of inputs for sexual and reproductive health, fast diagnosis to secure this spirochete, and regular and effective treatment of syphilis should be planned and executed. It is also essential to create and develop an educational program for the care and social inclusion of FSWs in cities in northern Brazil. Key populations must be valued and embraced through strategies that promote health and well-being. All of these pathways will contribute to reducing the spread of this spirochete among FSWs.

## Figures and Tables

**Figure 1 pathogens-10-00923-f001:**
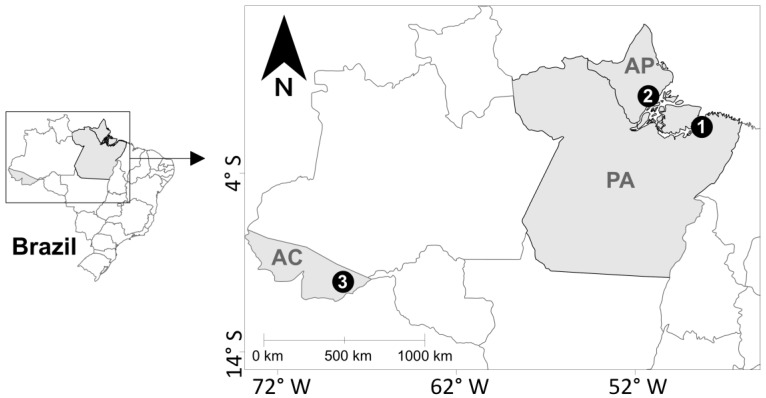
Geographic location of the cities where biological samples and personal information were collected from female sex workers in northern Brazil, states of Acre (AC), Amapá (AP) and Pará (PA). Points = cities: (1) Belém, (2) Macapá, and (3) Rio Branco.

**Table 1 pathogens-10-00923-t001:** Characteristics of the sample of female sex workers from the main cities in the states of Acre (Rio Branco), Amapá (Macapá), and Pará (Belém), northern Brazil.

Characteristics	Belém (N = 360)	Macapá (N = 31)	Rio Branco (N = 24)	All (N = 415)
	*n* (%)	*n* (%)	*n* (%)	*n* (%)
Age (years)				
Mean	43.0	30.5	25.1	35.1
Range	15–71	15–46	14–36	14–71
Marital status				
Single	276 (76.7)	17 (54.8)	20 (83.4)	313 (75.4)
Married	58 (16.1)	10 (32.2)	2 (8.3)	70 (16.9)
Divorced/widowed	26 (7.2)	4 (12.9)	2 (8.3)	32 (7.7)
School years				
≤8	258 (71.7)	22 (71.0)	17 (70.8)	297 (71.6)
>8	102 (28.3)	9 (29.0)	7 (29.2)	118 (28.4)
Monthly income				
Up to one wage *	256 (71.1)	23 (74.2)	15 (62.5)	294 (70.8)
More than one wage	102 (28.9)	8 (25.8)	9 (37.5)	121 (29.2)
Use of illicit drugs				
Yes	202 (56.1)	16 (51.6)	14 (58.3)	232 (55.9)
No	158 (43.9)	15 (48.4)	10 (41.7)	183 (44.1)
Condom use **				
Yes	235 (65.3)	18 (58.0)	16 (66.7)	269 (64.8)
No	10 (2.8)	3 (9.7)	2 (8.3)	15 (3.6)
Sometimes	115 (31.9)	10 (32.3)	6 (25.0)	131 (31.6)
Anal sex **				
Yes (often/sometimes)	106 (29.4)	12 (38.7)	8 (33.3)	126 (30.4)
No	254 (70.6)	19 (61.3)	16 (66.7)	289 (69.6)
Clients per week **				
Average	10.3	7.8	11.5	9.5
Range	6–24	5–23	4–40	5–40
Clients from other states of Brazil				
Yes	220 (61.1)	10 (32.3)	19 (79.2)	249 (60.0)
No	84 (23.3)	8 (25.8)	3 (12.5)	95 (22.9)
Do not know	56 (15.6)	13 (41.9)	2 (8.3)	71 (17.1)
Clients from other countries				
Yes	121 (33.6)	8 (25.8)	17 (70.8)	146 (35.2)
No	153 (42.5)	16 (51.6)	5 (20.9)	174 (41.9)
Do not know	86 (23.9)	7 (22.6)	2 (8.3)	95 (22.9)
STI history ^≠^				
Yes	80 (22.2)	8 (25.8)	8 (33.3)	96 (23.1)
No	280 (77.8)	23 (74.2)	16 (66.7)	319 (76.9)
Exposure to *T. pallidum*				
Recent infection	60 (16.7)	4 (12.9)	1 (4.2)	65 (15.7)
Past infection	71 (19.7)	9 (29.0)	5 (20.8)	85 (20.5)

*Average of the Brazilian minimum wage from 2009 to 2010 = R$ 465.00 (equivalent to US$ 103.00). ** In the last 30 days. ^≠^ STI: sexually transmitted infection.

**Table 2 pathogens-10-00923-t002:** Results of serological tests for *Treponema pallidum* in a sample of female sex workers in northern Brazil.

Rapid Plasma Reagin *	Enzyme-Linked Immunosorbent Assay **	Number/Total (%; 95% CI)
Nonreactive	Negative	256/415 (61.6; 58.0–65.3)
Reactive	Negative	9/415 (2.2; 0.0–4.8)
Reactive	Positive	65/415 (15.7; 12.3–19.4)
Nonreactive	Positive	85/415 (20.5; 17.4–24.4)
Nonreactive/reactive	Positive	150/415 (36.1; 33.2–39.8)

* Nontreponemal screening test, ** treponemal test; 95% CI: 95% confidence interval.

**Table 3 pathogens-10-00923-t003:** Factors associated with exposure to *Treponema pallidum* among female sex workers (FSWs) in northern Brazil using bivariate and multivariate analyses.

Factors	Total Number of FSWs	Number of FSWs Exposed	Bivariate OR (95% CI)	Multivariate aOR (95% CI)
Up to 8 years of study	297	123	2.4 (1.3–3.7)	2.9 (1.6–4.8)
Up to one minimum wage per month	294	129	3.6 (2.2–6.3)	4.1 (1.7–7.8)
Use of illicit drugs	232	108	2.9 (1.8–4.6)	3.7 (2.0–5.5)
Unprotected sex (inconsistent condom use) *	146	88	5.1 (3.1–7.9)	6.4 (3.2–9.0)
Anal sex (often + sometimes) *	126	77	4.3 (2.4–6.8)	4.6 (2.8–7.7)
More than 10 sexual partners per week *	287	121	2.5 (1.5–4.1)	3.3 (2.1–5.2)
STI history	96	67	6.6 (3.9–10.7)	7.1 (4.1–11.8)

* In the last 30 days. OR: odds ratio; aOR: adjusted OR; 95% CI: 95% confidence intervals.

## Data Availability

The data analyzed in the current study are not publicly available due to the progress of the analyses of possible infections and coinfections with other pathogens but are available from the corresponding author on request.

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
