# Peer review of "Treponema pallidum among Female Sex Workers: A Cross-Sectional Study Conducted in Three Major Cities in Northern Brazil"

_pathogens, 2021, doi:10.3390/pathogens10080923_

Round 1

Reviewer 1 Report

This is a well written manuscript.  I have a few comments & queries.

  1. Line 45.  Is this an annual occurrence of new syphilis cases globally?
  2. Line 49. You have used both a full stop & a comma in the same line to denote a decimal point.  Should be consistent throughout.
  3. Line 99 ( & in a few other places) T. pallidum should be italicised throughout.
  4. Table 2. In the column under rapid plasma reagin you use the terms "Non-reagent" & "reagent". Do you mean  "non-reactive" and "reactive"?
  5. Laboratory tests line 223. You have defined your interpretation of the serological results (shown in table 2). However you should perhaps consider alternate interpretations and perhaps discuss these.  See below.

For example 20% of primary syphilis cases may have a negative RPR test but positive on treponemal antibody tests. (e.g. Towns et al STI 2016 Vol 92, issue 2, pp 110 - 115 "Painful and multiple anogenital lesions are common in men with Treponema pallidum PCR-positive primary syphilis without herpes simplex virus coinfection: a cross-sectional clinic-based study")

Individuals may also remain serofast (ie a serological non-responder) after treatment, so a RPR of 8 does not necessarily represent an active infection. (e.g. Seña et al. BMC Infectious Diseases (2015) 15:479 "A systematic review of syphilis serological treatment outcomes in HIV-infected and HIV-uninfected persons: rethinking the significance of serological non-responsiveness and the serofast state after therapy" )

Author Response

We would like to thank all the reviewer's comments for the improvement of the manuscript and for the time available.

  1. Line 45.  Is this an annual occurrence of new syphilis cases globally?

Author’s response: Yes. It refers to the number of cases estimated worldwide

  1. Line 49. You have used both a full stop & a comma in the same line to denote a decimal point.  Should be consistent throughout.

Author’s response: We appreciate the comment and correct the text.

  1. Line 99 ( & in a few other places)  pallidum should be italicised throughout.

Author’s response: The species name has been corrected throughout the text.

  1. Table 2. In the column under rapid plasma reagin you use the terms "Non-reagent" & "reagent". Do you mean  "non-reactive" and "reactive"?

Author’s response: We thank you for your note and have changed the terms to Non-reactive and Reactive

  1. Laboratory tests line 223. You have defined your interpretation of the serological results (shown in table 2). However you should perhaps consider alternate interpretations and perhaps discuss these.  See below.

For example 20% of primary syphilis cases may have a negative RPR test but positive on treponemal antibody tests. (e.g. Towns et al STI 2016 Vol 92, issue 2, pp 110 - 115 "Painful and multiple anogenital lesions are common in men with Treponema pallidum PCR-positive primary syphilis without herpes simplex virus coinfection: a cross-sectional clinic-based study")

Individuals may also remain serofast (ie a serological non-responder) after treatment, so a RPR of 8 does not necessarily represent an active infection. (e.g. Seña et al. BMC Infectious Diseases (2015) 15:479 "A systematic review of syphilis serological treatment outcomes in HIV-infected and HIV-uninfected persons: rethinking the significance of serological non-responsiveness and the serofast state after therapy" )

Author’s response: We greatly appreciate the observation and agree that the interpretation of serology for immunological diagnosis of syphilis is complex and should be associated with the clinical examination and history of each patient. We used one of the algorithms established by the Ministry of Health of Brazil for the diagnosis of syphilis and added the information provided by the reviewer in results and discussion, as shown below:

Table 2:  

Rapid Plasma Reagin*     Enzyme-linked immunosorbent assay**                   Interpretation

Non- reactive                                         Negative                        Susceptible/Incubating syphilis

Reactive                                                  Negative                        Susceptible/False positive RPR

Reactive                                                   Positive                         Recent or past infection

Non- reactive                                         Positive                         Early syphilis/Past infection

Non- reactive/Reactive                         Positive                         Exposure to T. pallidum

In discussion:

“Syphilis serology results are not always easy to interpret given the complex patho-genesis of the infection and the immune response of each individual. In the present study, we considered active syphilis (ongoing or recent infection) when the RPR result was ≥ 1:8, however, it is noteworthy that about 20% of cases of primary syphilis may have a negative RPR test, in the presence of treponemal antibodies [17]. Regardless of any initial interpretation, in the case of suspected syphilis, all participants were properly treated. In addition, it may occur from individuals manifesting less than four-fold de-cline or persistently positive RPR titers (serofast), which may represent treatment failure, reinfection or a satisfactory immune response [18].”

Reviewer 2 Report

This is a well written and interesting paper and certainly an important topic.

What is the N in this table? I had some difficulty interpreting Table 3.  Is it based on the total of 415 respondents?  I think the table requires clarification.

Why is the prevalence in  this area of Brazil so high? You suggest poor education, low income, trade and tourism, but there are other low income rural areas that do not report a similar prevalence.  You seem to imply on line 188 that it may be due to many refugees from Haiti & Venezuela.  But, You may remember that for a while the USA blamed the growing prevalence of HIV in the USA on refugees from Haiti, which turned out not to be true. I can think of other possibilities.  Maybe that is an area for further research.

The conclusion section seems more like a discussion section where you discuss your results.  You might conclude by pointing to what further research should be done to expand upon your  findings? What do you suggest can be done now to meet the urgent need you point to in the last sentence of the conclusions.

Author Response

We would like to thank all the reviewer's comments for the improvement of the manuscript and for the time available.

What is the N in this table? I had some difficulty interpreting Table 3.  Is it based on the total of 415 respondents?  I think the table requires clarification.

Author’s response: The authors modified table 3 in order to facilitate the interpretation of the findings. In this table, the first column lists the factors associated with exposure to Treponema pallidum; the second column shows how many female sex workers (FSWs) performed this factor; the third column how many FSWs that performed this factor were also exposed to T. pallidum; columns 4 and 5 indicate the values of the bivariate and multivariate analyses, respectively, for each factor. See new table 3.

Why is the prevalence in this area of Brazil so high? You suggest poor education, low income, trade and tourism, but there are other low income rural areas that do not report a similar prevalence.  You seem to imply on line 188 that it may be due to many refugees from Haiti & Venezuela.  But, You may remember that for a while the USA blamed the growing prevalence of HIV in the USA on refugees from Haiti, which turned out not to be true. I can think of other possibilities.  Maybe that is an area for further research.

Author’s response: Currently, the authors are conducting studies on the perception of the risk of acquiring STI, the barriers to accessing existing health services, and the violence and discrimination experienced by FSWs. Findings from these new studies will help to better understand the epidemiological scenario and suggest safer and more effective interventions. This fact was included in the manuscript (“These two important points, the perception of risk to STI and the barriers to accessing existing health services, should be studied in the future. Overall, these interventions and complementary studies are essential to understand the epidemiological scenario and reduce the spread of T. pallidumand other pathogens among FSWs and, consequently, in the general population”).

The conclusion section seems more like a discussion section where you discuss your results.  You might conclude by pointing to what further research should be done to expand upon your  findings? What do you suggest can be done now to meet the urgent need you point to in the last sentence of the conclusions.

Author’s response: The authors modified the text referring to the study conclusions. Actions related to diagnosis, treatment and prevention were highlighted, as well as the importance of creating and running a program to seek, welcome, care for and interact with these vulnerable women and, from there, discover more information on the epidemiological scenario. New text in manuscript: “High rates of infections (past and recent) by T. pallidum were detected among FSWs in the cities of Belém, Macapá, and Rio Branco. Actions related to the supply of inputs for sexual and reproductive health, diagnosis fast and secure this spirochete, regular and effective treatment of syphilis should be planned and executed. It is also essential to create and develop an educational program for the care and social inclusion of FSWs in cities in northern Brazil. Key populations must be valued and embraced by strategies that promote health and well-being. All of these pathways will contribute to reducing the spread of this spirochete among FSWs”.
